# NETWORK ROBUSTNESS TO PCA PERTURBATIONS

## ABSTRACT

A key challenge in analyzing neural networks' robustness is identifying input features for which networks are robust to perturbations. Existing work focuses on direct perturbations to the inputs, thereby studies network robustness to the lowest-level features. In this work, we take a new approach and study the robustness of networks to the inputs' semantic features. We show a black-box approach to determine features for which a network is robust or weak. We leverage these features to obtain provably robust neighborhoods defined using robust features and adversarial examples defined by perturbing weak features. We evaluate our approach with PCA features. We show (1) our provably robust neighborhoods are larger: on average by 1.5x and up to 4.5x, compared to the standard neighborhoods, and (2) our adversarial examples are generated using at least 12.2x fewer queries and have at least 2.8x lower $L_2$ distortion compared to state-of-the-art. We further show that our attack is effective even against ensemble adversarial training.

## 1 INTRODUCTION

The reliability of deep neural networks (DNNs) has been undermined by adversarial examples: small perturbations to inputs that deceive the network (e.g., Goodfellow et al. (2015)). A key step in recovering DNN reliability is identifying input features for which the network is robust. Existing work focuses on the input values, the lowest-level features, to evaluate the network robustness. For example, a lot of work analyzes networks' robustness to neighborhoods consisting of all inputs at a certain distance from a given input (e.g., Boopathy et al. (2019); Katz et al. (2017); Salman et al. (2019); Singh et al. (2019a); Tjeng et al. (2019); Wang et al. (2018)). Despite the variety of approaches introduced to analyze robustness, the diameter $\epsilon$ (controlling the neighborhood size) of the provably robust neighborhoods is often very small. This may suggest an inherent barrier of the robustness of DNNs to distance-based neighborhoods. To illustrate, consider Figure 1(a) and Figure 1(b) – which are visibly the same but in fact each of their pixels differs by $\epsilon = 0.026$. That $\epsilon$ is the maximal one for which the $L_\infty$ ball $B_\epsilon(x)$ ($x$ is Figure 1(a)) was proven robust by ERAN (Singh et al., 2018; Gehr et al., 2018), a state-of-the-art robustness analyzer.

**Feature-defined neighborhoods** We propose to analyze network robustness to *perturbations of high-level input features*. A small perturbation to a feature translates to changes of multiple input entries (e.g., image pixels) and as such may produce visible perturbations. To illustrate, consider a neighborhood around Figure 1(a) in which only the background pixels can change their color. It turns out that, for this neighborhood, ERAN – the same robustness analyzer – is able to prove a neighborhood which has $10^{672}$x more images. Figure 1(c) shows a maximally perturbed image in this neighborhood, and Figure 1(d) illustrates two other images in it. These images are visibly different from Figure 1(a). Proving such neighborhood robust, for many inputs, can suggest that the network is robust to background color perturbations, thereby provide insights to the patterns the network learned.

**Key idea: robust features** An inherent challenge in finding robust feature-defined neighborhoods is automatically finding good candidate features (e.g., background color). Part of this challenge stems from the substantial running time of any robustness analyzer on a *single* neighborhood. This makes brute-force search of feature-defined neighborhoods for a large number of features and inputs futile. We propose a sampling approach to identify features which are likely to be robust for many inputs. We call these *robust features*. We experimentally observe that our robust features generalize to unseen inputs, even though they were determined from a (small) set of inputs.

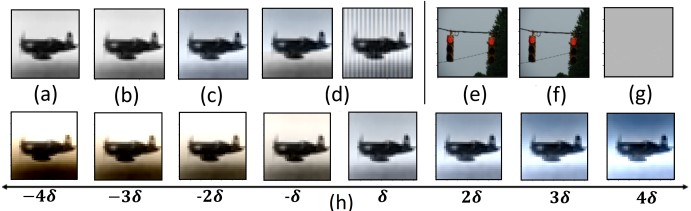

Figure 1: Network robustness to feature perturbations. (a) An image $x$ from CIFAR10. (b) A maximally perturbed input from the maximally $L_\infty$ neighborhood $B_\epsilon(x)$ proven robust by ERAN. (c) A maximally perturbed input (sky color is blue instead of gray) from the maximally feature-defined neighborhood $N_\delta^{\mathtt{sky\_color}}(x)$ proven robust by ERAN. (d) Other images in $N_\delta^{\mathtt{sky\_color}}(x)$. (e) A traffic light image from Imagenet, correctly classified by Inception-V3. (f) Our feature-based adversarial example classified by the same model as crane. (g) A visualization of the difference. (h) Perturbations to the image $x$ using the second PCA feature (whose semantic meaning is sky color).

**Feature-guided adversarial examples**  Dually to robust features, we define and identify *weak features*. We show how to exploit them in a simple yet effective black-box attack. Our attack perturbs a given input based on the weak features and then greedily reduces the number of modified pixels to obtain an adversarial example which minimizes the $L_2$ distance from the original input. Figure 1 illustrates our attack on ImageNet with the Inception-V3 architecture (Szegedy et al., 2016). Figure 1(e) shows a traffic light image (correctly classified), Figure 1(f) shows our feature-guided adversarial example (classified as crane), and Figure 1(g) visualizes the difference between the two images. We experimentally show that our attack is competitive with state-of-the-art practical black-box attacks (AutoZoom by Tu et al. (2019) and GenAttack by Alzantot et al. (2019)) and fools the ensemble adversarial training defense by Tramèr et al. (2018). Our results strengthen the claim of Ilyas et al. (2019) who suggested that weak (non-robust) features of the data contribute to the existence of adversarial examples. However, unlike Ilyas et al. (2019), we do not focus on features that stem from the DNN. This allows us to expose interesting patterns, expressed via simple functions, that the DNN has learned or missed.

**PCA features**  We obtain an initial set of features by running *principal component analysis* (PCA). PCA provides us with an automatic way to extract useful features from a dataset. For example, the sky color feature of the airplane in Figure 1(a) is in fact the second PCA dimension of this class of images in CIFAR10. Figure 1(h) shows the effect of perturbing this feature by small constants (multiplied by $\delta = 5$) in the PCA domain and projecting it back to the image domain. Our choice of using PCA is inspired by earlier work. We hypothesized that DNN may learn (some) of the PCA features by relying on works that showed that PCA can capture semantic features of the dataset (e.g., Zhoua et al. (2008); Jolliffe (2002)) and that DNNs learn semantic features (e.g., Zeiler & Fergus (2014)). Further, PCA has been linked to adversarial examples: several works showed how to detect adversarial examples using PCA (Hendrycks & Gimpel., 2017; Jere et al., 2019; Li & Li, 2017), others utilized PCA to approximate an adversarial manifold (Zhang et al., 2020), and others constructed an attack that modified only the first PCA dimensions (Carlini & Wagner., 2017). Computing the exact PCA dimensions requires perfect-knowledge of the dataset and is time-consuming for large datasets (e.g., ImageNet). We show that an approximation of the PCA dimensions is sufficient to obtain robust and weak features and that these can be computed from a small subset of the dataset (not used for training). The assumption that the attacker has access to a small dataset of similar distribution to the training set is often valid in practice (e.g., for traffic sign recognition benchmarks, like GTSRB, or face recognition applications).

We evaluate our approach on six datasets and various network architectures. Results indicate that (1) compared to the standard neighborhoods, our provably robust feature-guided neighborhoods have larger volume, on average by 1.5x and up to 4.5x, and they contain $10^{79}$x more images (average is taken over the exponent), and (2) our adversarial examples are generated using at least 12.2x fewer queries and have at least 2.8x lower $L_2$ distortion compared to state-of-the-art practical black-box attacks. We also show that our attack is effective even against ensemble adversarial training.

To conclude, our main contributions are:

- Definitions of networks' robustness and weakness to feature perturbations (Section 2).
- A sampling algorithm for finding the most robust and weakest features (Section 3).
- Feature-based neighborhoods and their robustness analysis (Section 4).
- Feature-based adversarial examples (Section 5).
- Instantiation with PCA features (Section 6).
- Extensive evaluation of our approach (Section 7).

## 2 ROBUST AND WEAK FEATURES

In this section, we define robust and weak features. We focus on classifiers, $D : \mathbb{R}^d \to \mathbb{R}^c$, given as a black-box, and trained on input-output pairs $(x, z)$ sampled from a distribution $\mathcal{T}$. We denote by $\text{class}(D(x))$ the classification of $x$ by $D$ (i.e., the index $i$ maximizing $D(x)_i$).

**Features**  A feature function $f : \mathbb{R}^d \to \mathbb{R}^d$ is a bijective function mapping inputs to real-valued vectors. The assumption that $f$ is bijective simplifies our definitions but these can be extended to other kinds of feature functions. The $i^{\text{th}}$ component of a feature function $f$ is called the feature $i$ of $f$.

**Feature perturbation**  For a feature function $f : \mathbb{R}^d \to \mathbb{R}^d$, index $i$, and a scalar $\delta \in \mathbb{R}$, we define the perturbation of $x$ along feature $i$ by $\delta$ as follows: $x_{f,i,\delta} = f^{-1}(f(x) + \delta_{\{i\}})$, where $\delta_{\{i\}} \in \mathbb{R}^d$ is a vector whose entries are zero except for the $i^{\text{th}}$ entry which is $\delta$.

**Network robustness to features perturbations**  Given a feature function $f$, an index $i$, and a scalar $\delta \in \mathbb{R}^+$, we say the feature $i$ of $f$ is $\delta$-robust if any perturbation to the feature $i$ up to $\delta$ does not decrease the *class confidence of $D$ on $x$* denoted by $CC(D, x)$ (shortly defined). Formally:

$$\mathbb{P}_{(x,z) \sim \mathcal{T}}([\forall \delta' \in [0, \delta].class(D(x_{f,i,\delta'})) = class(D(x)) \text{ and } CC(D, x_{f,i,\delta'}) \geq CC(D, x)]) = 1.$$

The expression $[\cdot]$ is the Iverson bracket: it is $1$ if the formula evaluates to true, and $0$ otherwise. We define $\delta$-robustness for $\delta \in \mathbb{R}^-$ similarly (except that $\delta' \in [\delta, 0]$). If $i$ is $\delta$-robust and $\delta > 0$, we say the robustness direction of $i$ is $+$ (positive); otherwise if $\delta < 0$ the direction is $-$ (negative).

We define network weakness dually. The feature $i$ of $f$ is $\delta$-weak if any perturbation to the feature $i$ up to $\delta$ changes the classification or decreases the class confidence. Formally, for $\delta \in \mathbb{R}^+$:

$$\mathbb{P}_{(x,z) \sim \mathcal{T}}([\forall \delta' \in [0, \delta].class(D(x_{f,i,\delta'})) \neq class(D(x)) \text{ or } CC(D, x_{f,i,\delta'}) \leq CC(D, x)]) = 1.$$

**Class confidence**  The definition of class confidence $CC(D, x)$ is orthogonal to our approach. One can define it as the class probability. However, we experimentally observed that it is better to relax the requirement about the absolute value and instead focus on the *difference* between the probability of the highest ranked class and the second highest, that is:

$$CC(D, x) = D(x)_{\text{class}(D(x))} - \max\{D(x)_i \mid i \in \{0, \dots, c-1\} \setminus \{\text{class}(D(x))\}\}.$$

**Most robust and weakest features**  Generally, we do not expect to find robust features if the features are defined orthogonally to the network's structure and parameters. However, if we relax the definition and quantify the *robustness level* of a feature, we can expose connections between the feature and the network's robustness behavior. We define the robustness level $RL$ as:

$$RL = \mathbb{P}_{(x,z) \sim \mathcal{T}}([\forall \delta' \in [0, \delta].class(D(x_{f,i,\delta'})) = class(D(x)) \text{ and } CC(D, x_{f,i,\delta'}) \geq CC(D, x)]).$$

We define the problem of finding the most robust features as finding $k$ features maximizing the robustness level. Dually, we define the weakness level and the problem of finding $k$ weakest features.

## 3 MOST ROBUST AND WEAKEST FEATURES VIA SAMPLING

In this section, we present our approach to approximate the most robust and weakest features via sampling. We approximate these features by checking robustness over inputs from a set $S$ (drawn from the distribution $\mathcal{T}$) and by sampling the intervals $[0, \delta]$ and $[-\delta, 0]$ at several points determined by a step parameter $\eta$. Algorithm 1 depicts our computation. The time complexity of Algorithm 1 is

---

**Algorithm 1:** FindRobustWeakFeatures($D$, $S$, $f$, $d$, $k$, $\delta$, $\eta$)

---

**Input:** A classifier $D$, input-output pairs $S$, a feature function $f$, a number of features $d$, a
number $k$, scalars $\delta$ and $\eta$ (precondition: $0 < \eta \leq \delta$ and $k \leq d$ ).
**Output:** $k$ most robust features and $k$ weakest features along with their direction.
$RL = [0, \ldots, 0]; WL = [0, \ldots, 0];$
**for** $i \in [0, \ldots, d-1]$ **do**
    **for** $(x, z) \in S$ **do**
        r = true; w = true;
        **for** $\delta' = \eta; \delta' \leq \delta; \delta' + = \eta$ **do**
            **if** $class(D(x_{f,i,\delta'})) \neq class(D(x))$ or $CC(D, x_{f,i,\delta'}) < CC(D, x)$ **then** r=false;
            **else** w=false;
        **if** $r$ **then** $RL[(i, +)] = RL[(i, +)] + 1;$
        **if** $w$ **then** $WL[(i, +)] = WL[(i, +)] + 1;$
        // a similar check for the negative direction $-$ (with $-\eta$ and $-\delta$) to update the key $(i, -)$

---

**return** *The keys (index-direction pairs) of the $k$ maximal entries in $RL$ and $WL$*

---

linear in $d \cdot |S| \cdot \lceil \delta/\eta \rceil \cdot (T_p + T_D)$, where $d$ is the number of features, $T_p$ the time to compute the feature perturbation $x_{f,i,\delta'}$ and $T_D$ the time to run $x_{f,i,\delta'}$ through the classifier $D$. The number of queries to the network is at most $2 \cdot d \cdot |S| \cdot \lceil \delta/\eta \rceil + |S|$. The term $|S|$ counts the number of queries for computing $class(D(x))$ and $CC(D, x)$ for every $(x, z) \in S$, computed once at the beginning of Algorithm 1 (this step is omitted from Algorithm 1 for simplicity's sake).

Our algorithm supports a trade-off between precision and performance: the larger the $S$ and the smaller the $\eta$, the more precise the result, but with a higher number of queries to the network. In practice, even though Algorithm 1 is executed once per dataset (after which many neighborhoods can be analyzed for robustness or attacks), its running time can be high. Thus, in our experiments, we set $|S|$ to 100, $d$ to 1,000 (i.e., we do not scan all features), and $\eta = \delta$. Despite these substantial restrictions, we still observe good results in practice (see Section 7). This encourages us that even a coarse search for robust and weak features is sufficient to obtain larger provably robust neighborhoods and an adversarial example attack which is competitive with state-of-the-art attacks.

## 4  MAXIMALLY ROBUST FEATURE-GUIDED NEIGHBORHOODS

In this section, we define feature-guided neighborhoods and explain how we find maximally robust neighborhoods. Given an input $x$, a feature $i$ of $f$, and a scalar $\delta \in \mathbb{R}^+$, *the $(f, i, \delta)$-neighborhood of $x$* is the set $N_\delta^{f,i}(x) = \{x_{f,i,\delta'} \mid \delta' \in [0, \delta]\}$ (the definitions for $\delta \in \mathbb{R}^-$ are similar and thus omitted). Given a DNN $D$, we say that $N_\delta^{f,i}(x)$ is robust if $D$ classifies all inputs in $N_\delta^{f,i}(x)$ the same. Our goal is to find maximally robust $(f, i, \delta)$-neighborhoods for a given $x$ and feature $i$ of $f$ (i.e., we look for a maximal scalar $\delta \in \mathbb{R}^+$). To this end, we employ a binary search in a given range $[0, u^N]$ (where $u^N$ is a constant). In each iteration of the search, we reason about the robustness of the current neighborhood using an existing analyzer, as shortly described. We note that for linear feature functions (e.g., PCA), the binary search is optimal. We further note that although our definition is general for any feature, larger neighborhoods are likely to be defined using the (most) robust features.

To verify robustness of a given $(f, i, \delta)$-neighborhood, we translate it to an $L_\infty$-norm neighborhood. This neighborhood can be then analyzed using an existing local robust analyzer (e.g., Gehr et al. (2018); Katz et al. (2017); Singh et al. (2019b)). In $L_\infty$-norm neighborhoods, each input entry (e.g., a pixel) can be perturbed independently. Formally, given an input $x$ and lower and upper bounds for every $j^{\text{th}}$ entry $\epsilon_j^l, \epsilon_j^u$, the $L_\infty$-norm neighborhood is: $N(x) = \{x' \mid \forall j, x_j' \in [x - \epsilon_j^l, x + \epsilon_j^u]\}$. (A special case is $B_\epsilon$ neighborhoods, in which all bounds are $\epsilon$, i.e., for all $j$, $\epsilon_j^l = \epsilon_j^u = \epsilon$.) For linear feature functions (e.g., PCA), we translate $(f, i, \delta)$-neighborhoods to $L_\infty$-norm ones as follows:

$$N_{f,i,\delta}^\infty(x) = \{x' \mid \forall j, x_j' \in [\min(x_j, (x_{f,i,\delta})_j), \max(x_j, (x_{f,i,\delta})_j)]\}.$$

For other feature functions, one has to obtain the minimal and maximal values of each input entry.

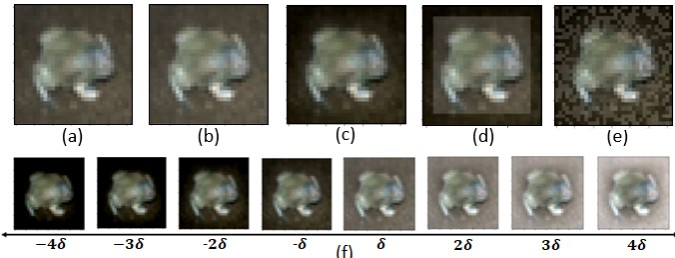

Figure 2: (a) A frog $x$ from CIFAR10. (b) A maximally perturbed frog from the $B_{\epsilon=0.024}(x)$ neighborhood – the maximal $\epsilon$ proven robust. (c) A maximally perturbed frog from the $(PCA, 0, \delta)$-neighborhood ($\delta = -6.66$). (d)+(e) Two other images in the $(PCA, 0, \delta)$-neighborhood of $x$. (f) Perturbations to $x$ using the PCA feature 0 (whose semantic meaning is background lighting level).

## 5 FEATURE-GUIDED ADVERSARIAL EXAMPLES

In this section, we present our feature-guided attack which perturbs the weakest features. Our attack consists of two steps: perturbing the weakest features by a small $\delta$ and then greedily refining the adversarial example. In addition to the expected parameters (the classifier $D$, the input-output pair $(x, z)$, and the feature function $f$), the attack takes as input the set of weak features $W$ consisting of indices and directions, a step scalar $\delta_s^{adv}$, and a maximal distortion scalar $\delta^M$. The first step of the attack looks for the smallest $\delta \leq \delta^M$, with granularity $\delta_s^{adv}$, for which $x_{f,W,\delta'}$ is an adversarial example. Here, $x_{f,W,\delta'} = f^{-1}(f(x) + \delta_W)$, where $\delta_W \in \mathbb{R}^d$ is a vector whose entries are zero except for indices from $W$ whose entries are $\delta$ or $-\delta$ depending on their direction. If an adversarial example is found, the second step of the attack refines it by greedily attempting to recover pixels to their original value in $x$, while still guaranteeing that the example is misclassified by $D$. To reduce the number of queries to $D$ and converge faster to a minimal adversarial example, the attack attempts to recover groups of adjacent pixels (e.g., of size $50$) together. If it fails, it cuts the group size in half. In practice, we observe that this approach provides a good balance between reducing the number of queries and recovering many input entries to their original value.

## 6 INSTANTIATION WITH PCA

In this section, we explain how we use PCA to instantiate our approach. We begin with a brief background, then present our instantiation and exemplify PCA-guided neighborhoods and attacks.

**Background** PCA projects inputs into an (interesting) set of features (dimensions), each represents a different aspect of the data and captures a different amount of variance (information). Technically, PCA is a statistical framework that projects a random variable $x \in \mathbb{R}^d$ onto a random variable $y \in \mathbb{R}^d$, such that the components $\{y_i\}_{i=1}^d$ are uncorrelated and have maximal variance values. The set $\{y_i\}_{i=1}^d$ is defined by $y_i = u_i^T x \in \mathbb{R}$, such that $u_i \in \mathbb{R}^d$, where the $u_i$-s are *the principal components* of $x$. The principal components are computed with the goals of (1) maximizing the variance of the components $\{y_i\}_{i=1}^d$ and (2) keeping these components uncorrelated. In practice, instead of the random variable $x$, we consider a data matrix $X$, consisting of $n$ input samples. Given the principal components, we define the PCA transformation matrix by $U = [u_1, u_2, \ldots, u_d]$. We use $U$ to project the dataset into the PCA domain and vice versa: $Y = XU \in R^{n \times d}$, $X = YU^T \in R^{n \times d}$.

**PCA feature functions** We observed that PCA provides better features when computed for inputs of the same class. Thus, for each class $z$, we define a feature function $PCA_z(x) = x^T U_z$, where $U_z$ is the PCA transformation matrix computed from inputs with output $z$. As a result, Algorithm 1 is executed $c$ times, where $c$ is the number of classes. The running time of computing the PCA feature functions is $c \cdot min(|S|^3, d^3)$. In practice, we only compute the first $d = 1,000$ PCA dimensions. Combined with the small size of $S$ ($|S| = 100$ per class), the running time is a few minutes per class.

We next show an example of a $(PCA, i, \delta)$-neighborhood (Appendix A shows more examples). Figure 2(a) shows a frog image $x$ from CIFAR10. The maximal $\epsilon$ for which the $B_\epsilon(x)$ was proven

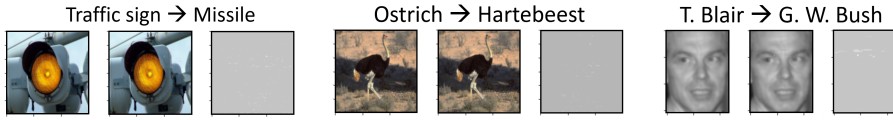

Figure 3: PCA-guided adversarial examples.

robust by ERAN (Gehr et al., 2018) was $\epsilon = 0.024$. Figure 2(b) shows a maximally perturbed image in this neighborhood. The maximal $\delta$ for which the $N_\delta^{PCA,0}(x)$ was proven robust by ERAN was $\delta = -6.66$ (the feature 0 represents lighting conditions). This PCA-guided neighborhood contains $10^{703}$x more images than the $B_\epsilon(x)$ neighborhood. Figure 2(c) shows a maximally perturbed image in this neighborhood, while Figure 2(d) and Figure 2(e) show two other images in it. Figure 2(f) illustrates the semantic meaning of the feature we perturb (background lighting conditions): it shows several images obtained by first perturbing this feature in the PCA domain by a small constant (a multiplication of $\delta = 5$) and then projecting the PCA perturbation back to the image domain.

Figure 3 shows examples of our PCA-guided attack (Appendix A has more examples). Each example shows a correctly classified input, our adversarial example, and the perturbation (scaled for better visualization). The leftmost example shows a traffic sign image from ImageNet whose adversarial example is classified as a missile by Inception-V3. The middle example shows an ostrich image from ImageNet whose adversarial example is classified as an hartebeest by the same model. The rightmost image shows an image of Blair whose adversarial example is classified by RESNET20 as Bush.

## 7 EVALUATION

In this section, we evaluate our approach and the applications from Sections 4 and 5.

**Setup**  We implemented our approach in Python using PyTorch (we also support Tensorflow models). We trained models and ran most experiments on a Tesla K80 GPU with 12GB RAM. Only for running ERAN (Gehr et al., 2018), we used a dual AMD EPYC 7742 server with 1TB RAM running Ubuntu 20.04.1. For computing the most robust and weakest features of a given class, we defined $S$ as a set of 100 random inputs of that class (which were not part of the training set), $d = 1,000$ (i.e., we considered only the first $1,000$ PCA features), $k = 250$, and $\delta = \eta = 1$. For $(f, i, \delta)$-neighborhoods, we set $u^N = 100$. For our attack, we set $\delta_s^{adv} = 0.1$ and $\delta^M = 10$. As an optimization, our attack iterates over $k \in \{50, 100, 250\}$ to reduce the distortion level. We evaluate our approach on six datasets (described in Appendix B). Each is split into a training set, a set for computing PCA+Algorithm 1, and a set to evaluate our neighborhoods and attack. Our attack was tested on state-of-the-art networks, and our robustness analysis was done on networks taken from ERAN's repository (denoted by [E]) and networks whose size is similar to the large networks from that repository. We trained most models, except for Inception-V3 for ImageNet (Szegedy et al., 2016) and the C&W models (Carlini & Wagner, 2017) for MNIST and CIFAR10 (and models marked with [E]).

**PCA + Algorithm 1 analysis**  For each dataset, we computed the first $d$ PCA features from the set $S$ containing 100 images. This step took less than two minutes for all datasets. Then, for each model, we computed the most robust and weakest features among these dimensions. This step is executed once, after which these features are used to define multiple neighborhoods and adversarial examples. This step is executed separately for each class and thus can be naturally parallelized. The average running time of Algorithm 1 was less than 90 seconds for most datasets and models, and at most ten minutes for ImageNet. The number of queries is $2 \cdot |S| \cdot d \cdot c = 2 \cdot 10^5 \cdot c$, where $c$ is the number of classes. While this number is not negligible, it is done as a preprocessing step (similarly to training a model as done by other attacks, e.g., Tu et al. (2019)). The amortized added queries of this step over the dataset ranges from a handful of queries (for ImageNet) to a few dozens (for MNIST).

**Feature-guided neighborhoods**  To evaluate robustness of $(f, i, \delta)$-neighborhoods, we ran several experiments. In each experiment, we fixed a model, a dataset, a class, and a robust feature $i$. We considered small models ($3\times100$ and $5\times250$), medium-sized models (LeNet5 with ~10,000 neurons), models that are considered big for robustness analysis (C&W with ~77,000 neurons), convolutional

Table 1: PCA-guided neighborhoods vs. $B_\epsilon$ neighborhoods: v is the neighborhood volume, #im the number of images in it, and $|u - l|^{\max}$ the maximal pixel range. Results are averaged over ten images. We provide a graphical interpretation of our robust features in Figure 4.

| Dataset | Model | Class | $i$ | $v_{PCA}$ | $v_\epsilon$ | $\#im_{PCA}$ | $\#im_\epsilon$ | $\|u - l\|^{\max}_{PCA}$ | $\|u - l\|^{\max}_\epsilon$ |
|---------|-------|-------|-----|-----------|--------------|--------------|-----------------|--------------------------|-----------------------------|
| MNIST | $3 \times 100^E$ | 2 | 1 | 21.0 | 20.8 | $10^{275}$ | $10^{620}$ | 0.247 | 0.046 |
| MNIST | ConvSmall$^E$ | 0 | 1 | 62.6 | 66.7 | $10^{388}$ | $10^{979}$ | 0.76 | 0.149 |
| MNIST | ConvMaxpool$^E$ | 9 | 0 | 1.7 | 2.9 | $10^{48}$ | $10^{46}$ | 0.019 | 0.006 |
| MNIST | VGG-based | 0 | 0 | 2.6 | 1.6 | $10^{70}$ | 1 | 0.026 | 0.003 |
| FMNIST | $5 \times 250$ | 1 | 1 | 8.7 | 7.5 | $10^{248}$ | $10^{245}$ | 0.063 | 0.016 |
| FMNIST | C&W | 2 | 7 | 3.2 | 2.4 | $10^{87}$ | 1 | 0.033 | 0.003 |
| LFW | LeNet5 | 2 | 9 | 9.4 | 6.6 | $10^{355}$ | $10^{46}$ | 0.038 | 0.008 |
| LFW | C&W | 0 | 2 | 6.4 | 4.5 | $10^{231}$ | 1 | 0.023 | 0.004 |
| LFW | C&W | 2 | 9 | 7.6 | 5.3 | $10^{283}$ | 1 | 0.031 | 0.005 |
| GTSRB | LeNet5 | 14 | 8 | 16.0 | 8.3 | $10^{481}$ | $10^{347}$ | 0.097 | 0.01 |
| GTSRB | C&W | 3 | 0 | 3.5 | 1.5 | $10^{255}$ | 1 | 0.017 | 0.002 |
| GTSRB | C&W | 15 | 3 | 3.6 | 0.8 | $10^{147}$ | 1 | 0.018 | 0.001 |
| CIFAR10 | LeNet5 | 9 | 1 | 5.5 | 4.3 | $10^{110}$ | 1 | 0.005 | 0.001 |
| CIFAR10 | ConvSmall$^E$ | 0 | 1 | 235.4 | 186.4 | $10^{3057}$ | $10^{2941}$ | 0.176 | 0.063 |
| CIFAR10 | ConvSmall$^E$ | 1 | 0 | 181.8 | 125.5 | $10^{2944}$ | $10^{2661}$ | 0.13 | 0.041 |
| CIFAR10 | ConvBig$^E$ | 3 | 5 | 92.9 | 64.7 | $10^{2085}$ | $10^{1919}$ | 0.089 | 0.021 |
| CIFAR10 | ConvBig$^E$ | 0 | 1 | 47.9 | 45.4 | $10^{1644}$ | $10^{1577}$ | 0.031 | 0.014 |
| CIFAR10 | C&W | 1 | 0 | 6.4 | 4.9 | $10^{96}$ | 1 | 0.005 | 0.002 |

Table 2: Our attack vs. AutoZOOM and C&W. $Q^b/Q$ is the number of queries to the network before/after postprocessing, and $L_2^b/L_2$ is the $L_2$ distance from the original image before/after postprocessing. We abbreviate RESNET with RES and Inception-V3 with IncV3.

| Dataset | Model | Our attack | | | | AutoZOOM | | | | C&W |
|---------|-------|-----------|---|---|---|----------|---|---|---|-----|
| | | $L_2$ $\times 10^{-3}$ | $Q$ $\times 10^3$ | $L_2^b$ $\times 10^{-3}$ | $Q^b$ | $L_2$ $\times 10^{-3}$ | $Q$ $\times 10^3$ | $L_2^b$ $\times 10^{-3}$ | $Q^b$ | $L_2$ $\times 10^{-3}$ |
| MNIST | C&W | 3.31 | 3.2 | 8.72 | 86 | 2.52 | 3.8 | 11.58 | 250 | 2.62 |
| FMNIST | VGG13 | 0.82 | 2.6 | 2.55 | 19 | 2.18 | 3.8 | 4.54 | 102 | 1.87 |
| LFW | RES20 | 0.40 | 0.9 | 1.82 | 9 | 1.60 | 3.8 | 3.70 | 56 | 0.16 |
| GTSRB | RES20 | 0.30 | 3.1 | 1.24 | 23 | 0.94 | 3.8 | 3.94 | 272 | 0.18 |
| CIFAR10 | C&W | 0.29 | 2.9 | 1.67 | 57 | 0.70 | 3.8 | 1.98 | 103 | 0.12 |
| CIFAR10 | RES50 | 0.27 | 3.4 | 1.11 | 14 | 0.71 | 3.8 | 2.06 | 70 | 0.13 |
| CIFAR10 | VGG16 | 0.28 | 3.3 | 1.69 | 32 | 1.52 | 3.8 | 7.75 | 39 | 0.20 |
| ImageNet | Inc-V3 | 0.01 | 217 | 0.16 | 42 | 0.016 | 250 | 0.66 | 2654 | 0.003 |

models of different sizes from ERAN's repository, and a VGG-based model that we trained (~141,000 neurons). We computed the maximally robust $(PCA, i, \delta)$-neighborhoods for the first ten images of that class. Table 1 reports the results: the average neighborhood volume (v$= \Sigma_j \epsilon_j^u - \epsilon_j^l$), the number of images in the neighborhood (shown as a power of ten: $10^e$; we take the average over the exponent; for average $e = 0$, we write 1), and the maximal pixel range (i.e., $max\{\epsilon_j^u - \epsilon_j^l \mid j\}$). We compare our approach with the standard approach of computing maximally robust $B_\epsilon(x)$ neighborhoods (using binary search and ERAN). Results indicate that (1) our neighborhoods have larger volumes: on average by 1.5x and up to 4.5x; (2) our neighborhoods have $10^{79}$x more images (average is over the exponent) and, for deep networks or complex datasets, often the maximal $B_\epsilon$ neighborhood contains a *single* image, and (3) our neighborhoods have larger maximal pixel range. Figure 4 illustrates the semantic meaning of the PCA features presented in Table 1. For each dataset, class, and feature, it shows three images: an initial image (middle) and two images obtained by perturbing the feature by $\pm\delta$ (for some $\delta$). The three images are ordered by the feature's robustness direction: if the direction is positive, then the rightmost image is the perturbation with respect to $+\delta$; otherwise, the rightmost image is the perturbation with respect to $-\delta$. The arrows in the figure show the robustness direction.

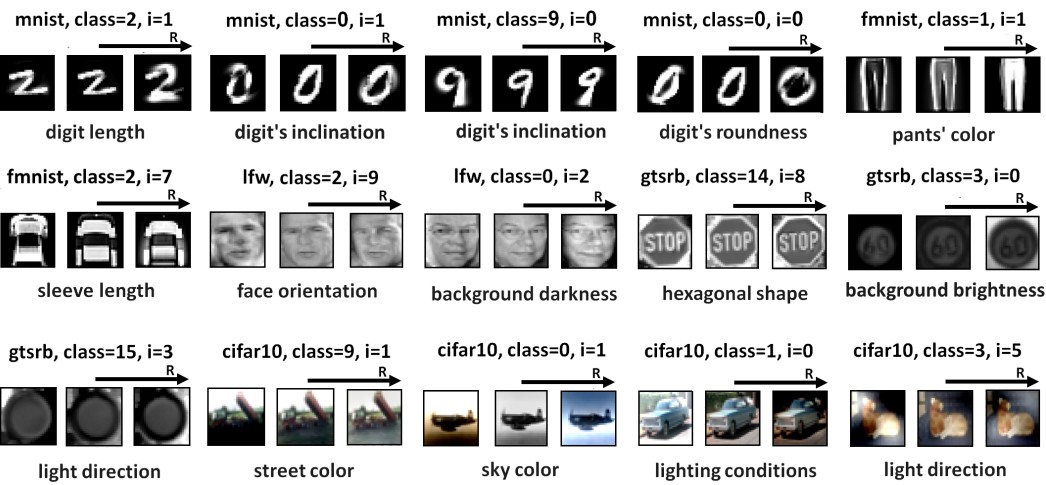

Figure 4: Illustration of the semantic meaning of the PCA features used in Table 1.

Table 3: Our attack vs. AutoZOOM and GenAttack on models trained with ensemble adversarial training. $S\%$ is the success rate and $L_2$ is the $L_2$ distance from the original image.

| Dataset | Our attack | | AutoZOOM | | GenAttack | |
|---|---|---|---|---|---|---|
| | $S\%$ | $L_2$ ($\times 10^{-3}$) | $S\%$ | $L_2$ ($\times 10^{-3}$) | $S\%$ | $L_2$ ($\times 10^{-3}$) |
| MNIST | 100 | 3.4 | 65 | 2.3 | 95 | 18.7 |
| F-MNIST | 100 | 0.8 | 100 | 2.1 | 94 | 11.9 |
| CIFAR10 | 100 | 0.3 | 100 | 1.3 | 100 | 5.1 |

**PCA-guided adversarial examples** We next evaluate our attack. In each experiment, we fix a dataset and model, and run our attack for 100 correctly classified inputs (not used during training or computing the most robust/weakest features). For each adversarial example, we measure the number of queries posed to the network and the $L_2$ distance from the original image. We measure these twice: (1) after perturbing the weakest features and obtaining the first adversarial example (denoted $Q^b$ and $L_2^b$) and (2) after the refinement postprocessing step (denoted $Q$ and $L_2$). We compare our approach with state-of-the-art practical black-box attacks: AutoZOOM by Tu et al. (2019) (which also has a refinement postprocessing step) and GenAttack by Alzantot et al. (2019) . We also compare our attack to the C&W attack Carlini & Wagner (2017), which is a white-box attack (with a full access to the model, including the internal layers – unlike our setting which assumes access only to the input and output layers). Table 2 shows the results for our attack, AutoZoom, and C&W, all of which have success rate of $100\%$. Results indicate that (1) our attack obtains the initial adversarial examples (before postprocessing) with 12.2x fewer queries and with 2.5x less $L_2$ distortion compared to AutoZOOM, (2) considering postprocessing, our attack generates adversarial examples with fewer queries and 2.8x less $L_2$ distortion compared to AutoZOOM, and (3) our black-box attack has 1.8x more $L_2$ distortion than the white-box attack of C&W (this is expected as a white-box setting provides the attacker with more knowledge). Due to lack of space, results for GenAttack are in Appendix C. Compared to GenAttack: (1) our attack obtains the initial adversarial examples with 18.7x fewer queries and with 4.1x less $L_2$ distortion, and (2) our final adversarial examples have 16.7x less $L_2$ distortion but with 29x more queries.

We next evaluate our attack for the same 100 inputs on models trained with ensemble adversarial training (Tramèr et al., 2018). To this end, we trained defended models for three datasets using trained RESNET50, RESNET20, and C&W models. We limited the query number to $10,000$. We compare our approach with AutoZoom and GenAttack. Table 3 shows the results. Results indicate that our attack has a higher success rate and in most cases it obtains smaller $L_2$ distortion.

In Appendix D, we provide experiments comparing our approach to an attack which uses random PCA features instead of the weakest features. These experiments show the advantage of our attack.

## 8 RELATED WORK

**Adversarial examples and PCA**   Several works consider PCA for finding and defending against adversarial examples. Some works use it to detect adversarial examples (Hendrycks & Gimpel., 2017; Jere et al., 2019; Li & Li, 2017). Bhagoji et al. (2018) train a model using the top-$k$ PCA components, while Carlini & Wagner. (2017) show that this approach is not immune to adversarial examples. These works focus on employing PCA as a detector of adversarial examples. In contrast, we use PCA as a feature extractor. Sanyal et al. (2020) show that encouraging DNNs to learn representations in lower-dimensional linear subspaces (including PCA) increases their robustness. Jetley et al. (2018) study classification quality versus vulnerability to adversarial attacks through principal directions and curvatures of the decision boundary. Zhang et al. (2020) use PCA to approximate an *adversarial region* which enables them to generate adversarial examples without a classifier. In contrast, we look for connections between features and classifiers' behavior.

**Feature analysis of DNNs**   Many works analyze a DNN with respect to features. Goswami et al. (2018) perturb face features manually, e.g., by covering the beard or eyes. Rozsa et al. (2019) focus on face images with pre-labeled features to study DNNs' robustness. Wicker et al. (2018) suggest using SIFT to extract images' local features and perturb them. Xie et al. (2019) suggest to denoise the features of the hidden CNN layers to increase robustness. In contrast, we take a black-box approach and automatically derive input features for which a DNN is robust. Ilyas et al. (2019) and Goh (2019) define robust and non-robust features (differently from us) and show that adversarial examples are linked to non-robust features. However, their features are inferred from the DNN, unlike our approach which considers features computed from the dataset. Further, we provide an approach to approximate the most robust and weakest features from a large set of features using a black-box access to the DNN.

**Adversarial examples**   Many adversarial example attacks were introduced. Some are white-box attacks with full access to the model, e.g., FGSM (Goodfellow et al., 2015), IFGSM (Kurakin et al., 2017), PGD (Madry et al., 2018), C&W (Carlini & Wagner, 2017), and EAD (Chen et al., 2018). While successful, all rely on the assumption that the attacker has full access to the model's parameters, which is often not true in practice. This encouraged others to study black-box attacks which only assume access to network's input and output layers (i.e., with no information on the internal layers). Papernot et al. (2017) show a black-box attack relying on a substitute technique, where the adversary trains a representative network. Such methods suffer from low success-rate because they rely on the assumption of adversarial example transferability, which does not always hold. Chen et al. (2017) present a black-box attack relying on gradient-estimation. This approach obtains high success-rate, however it is inefficient because it requires a high number of queries to the network. Recent works introduce practical and efficient black-box techniques: AutoZoom (Tu et al., 2019) uses efficient random gradient-estimation, while GenAttack (Alzantot et al., 2019) employs a genetic algorithm. In contrast to these approaches, our black-box attack depends on weak dataset's features.

**Local robustness**   Several works present approaches to analyze local robustness of DNNs. Anderson et al. (2019); Gehr et al. (2018); Ghorbal et al. (2009); Singh et al. (2018) rely on abstract interpretation, Katz et al. (2017) extend the simplex method, Boopathy et al. (2019); Salman et al. (2019) rely on linear relaxations, Dvijotham et al. (2018); Raghunathan et al. (2018) use duality, and Singh et al. (2019c); Wang et al. (2018) combine solvers with approximate methods. These works analyze the robustness of $L_\infty$-norm neighborhoods. Balunovic et al. (2019); Engstrom et al. (2019); Fawzi et al. (2017); Singh et al. (2019b) analyze robustness to geometric transformations (e.g., rotation, scaling). In contrast, we analyze robustness of neighborhoods defined by input features.

## 9 CONCLUSION

We presented the concept of network robustness and weakness to feature perturbations. We approximated the most robust and weakest features via sampling and leveraged them to define feature-guided neighborhoods and adversarial examples. Experimental results show that our provably robust feature-guided neighborhoods are much larger than the standard provably robust neighborhoods and that our adversarial examples require fewer queries and have lower $L_2$ distortion compared to state-of-the-art.

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

**F-MNIST**
**Model: $2 \times 100$-fully connected**
$\varepsilon = 0.045, \ \delta = 3.47, \ i = 1$
$|u - l|_{\text{PCA}}^{\text{avg}} = 0.18, |u - l|_{\text{PCA}}^{\text{max}} = 0.32$

**MNIST**
**Model: $2 \times 100$-fully connected**
$\varepsilon = 0.047, \ \delta = 9.62, \ i = 1$
$|u - l|_{\text{PCA}}^{\text{avg}} = 0.43, |u - l|_{\text{PCA}}^{\text{max}} = 1$

**GTSRB**
**Model: LeNet5**
$\varepsilon = 0.003, \ \delta = 5.44, \ i = 1$
$|u - l|_{\text{PCA}}^{\text{avg}} = 0.36, |u - l|_{\text{PCA}}^{\text{max}} = 0.46$

**(a)** **(b)** **(c)** **(d)** **(e)**

Figure 5: Examples of provably robust PCA-guided neighborhoods. (a) An input $x$. (b) A maximally perturbed input from the $B_\epsilon(x)$ neighborhood, for the maximal $\epsilon$ proven robust ($\epsilon$ is indicated to the left). (c) A maximally perturbed input from the $(PCA, 1, \delta)$-neighborhood of $x$ ($\delta$ is indicated to the left). (d)+(e) Two input examples in the $(PCA, 1, \delta)$-neighborhood of $x$.

## A   ADDITIONAL EXAMPLES

In this section, we show more examples of provably robust feature-guided neighborhoods and PCA-guided adversarial examples.

In Figure 5, we show for each dataset:

- an image;
- a maximally perturbed input from the $B_\epsilon(x)$ neighborhood, for the maximal $\epsilon$ that ERAN proved robust ($\epsilon$ is shown in the figure);
- a maximally perturbed input from the $(PCA, 1, \delta)$-neighborhood ($\delta$ is shown in the figure);
- two more input examples from the $(PCA, 1, \delta)$-neighborhood.

In the figure, we also indicate the average range of pixels (when translating the PCA features to the input domain) $|u - l|_{\text{PCA}}^{\text{avg}}$, and the maximal range of pixels $|u - l|_{\text{PCA}}^{\text{max}}$. In $B_\epsilon(x)$ neighborhoods, $|u - l|_\epsilon = 2 \cdot \epsilon$, for all pixels.

These examples demonstrate that (1) provably robust PCA-guided neighborhoods are larger than their corresponding $B_\epsilon(x)$ neighborhoods and (2) provably robust PCA-guided neighborhoods contain inputs which are *semantically* similar to the original image but are also visually different from it.

In Figure 6, we show PCA-guided adversarial examples, for ImageNet and an Inception-V3 model, computed on correctly classified inputs. The figure shows: the original image (left), the PCA-guided adversarial example (center), and the perturbation (right).

## B   DATASETS

In this section, we describe the datasets we used in our evaluation.

- ***MNIST:*** $28 \times 28$ images of 0-9 digits, ten classes.
- ***Fashion MNIST (F-MNIST):*** $28 \times 28$ pixel images of fashion items, ten classes.
- ***CIFAR10:*** $32 \times 32 \times 3$ pixel images, ten classes.

We imported these datasets from the PyTorch library. We also considered the following datasets:

- ***LFW sklearn version:*** $62 \times 47$ images of funneled faces, each classified as a person. In total, there are 5,749 different people (i.e., classes) and the dataset contains 13,233 images. We ignored classes with fewer than 100 samples, and thus remained with 1,140 samples and five classes (Colin Powell, Donald Rumsfeld, George W. Bush, Gerhard Schroeder, and Tony Blair). We imported this dataset from the *sklearn* library.

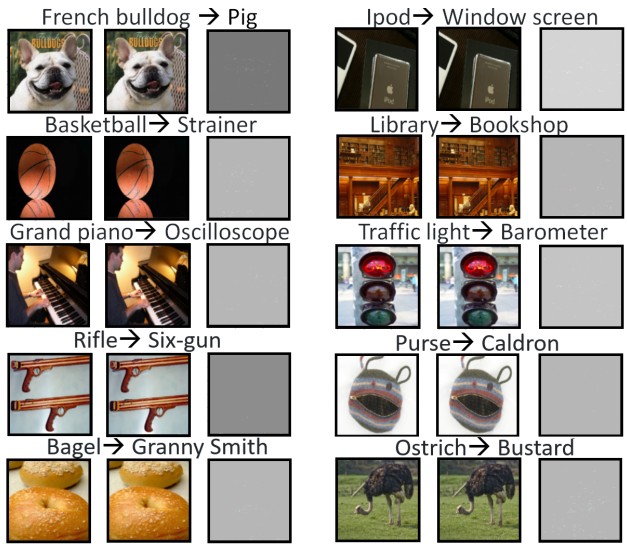

Figure 6: PCA-guided adversarial examples for ImageNet on an Inception-V3 model.

Table 4: Our attack vs. GenAttack. $S$ is success rate, $Q^b/Q$ is the number of queries before/after postprocessing, and $L_2^b/L_2$ is the $L_2$ distance from the original image before/after postprocessing.

| Dataset | Model | Our attack | | | | | GenAttack | | |
|---|---|---|---|---|---|---|---|---|---|
| | | S | $L_2$ $\times 10^{-3}$ | $Q$ $\times 10^3$ | $L_2^b$ $\times 10^{-3}$ | $Q^b$ | S | $L_2$ $\times 10^{-3}$ | $Q$ |
| MNIST | C&W | 1 | 3.31 | 3.2 | 8.72 | 86.25 | 1 | 13.8 | 71.4 |
| F-MNIST | VGG13 | 1 | 0.82 | 2.6 | 2.55 | 18.9 | 0.95 | 11.3 | 158.8 |
| LFW | RESNET20 | 1 | 0.40 | 0.9 | 1.82 | 8.8 | 1 | 11.2 | 236.4 |
| GTSRB | RESNET20 | 1 | 0.30 | 3.1 | 1.24 | 22.8 | 1 | 6.4 | 200.8 |
| CIFAR10 | C&W | 1 | 0.29 | 2.9 | 1.67 | 57.1 | 1 | 5.7 | 56.6 |
| CIFAR10 | RESNET50 | 1 | 0.27 | 3.4 | 1.11 | 14.1 | 1 | 5.5 | 37.2 |
| CIFAR10 | VGG16 | 1 | 0.28 | 3.3 | 1.69 | 31.6 | 0.97 | 5.8 | 402.8 |
| ImageNet | Inception-V3 | 1 | 0.11 | 5 | 0.16 | 42.3 | 1 | 0.66 | 3762.5 |

- ***The German Traffic Sign Recognition Benchmark (GTSRB):*** 60,000 images of 43 different traffic signs (taken from `http://benchmark.ini.rub.de/`). We cropped the images using the coordinates that the dataset provided, and resized to $32 \times 32$.
- ***ImageNet:*** $299 \times 299 \times 3$ images of 1,000 different objects. Our attack was evaluated on a small subset, taken from `http://www-personal.umich.edu/~timtu/Downloads/imagenet_npy/imagenet_test_data.npy/`, with images from 10 classes. We used a pretrained Inception-V3 model, taken from `http://jaina.cs.ucdavis.edu/datasets/adv/imagenet/inception_v3_2016_08_28_frozen.tar.gz`.

## C    COMPARISON TO GENATTACK

In this section, we compare our attack to GenAttack (Alzantot et al., 2019) on 100 correctly-classified inputs. Since GenAttack has no postprocessing step, we only report the number of queries $Q$ and the $L_2$ distortion. We limited the query number of both approaches to 5,000 (for lower number of queries, GenAttack has lower success rate). Table 4 shows the results. Results indicate that (1) our attack obtains the initial adversarial examples with 18.7x fewer queries and with 4.1x less $L_2$ distortion, and (2) our final adversarial examples have 16.7x less $L_2$ distortion but with 29x more queries.

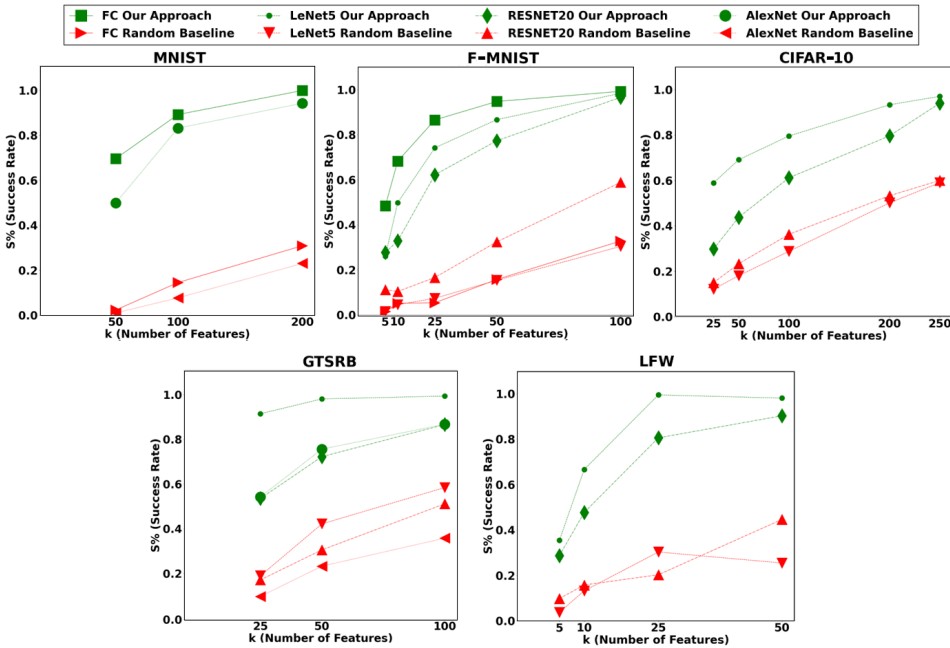

Figure 7: Success rate of PCA-guided adversarial examples vs. a random baseline as a function of $k$.

Table 5: Our $L_2$ distortion in the PCA domain.

| Dataset | Model | $L_2^{PCA}$ $\times 10^{-3}$ |
|---------|-------|------------------------------|
| MNIST | C&W | 3.35 |
| F-MNIST | VGG13 | 0.83 |
| LFW | RESNET20 | 0.28 |
| GTSRB | RESNET20 | 0.23 |
| CIFAR10 | C&W | 0.204 |
| CIFAR10 | RESNET50 | 0.201 |
| CIFAR10 | VGG16 | 0.206 |
| ImageNet | Inception-V3 | 0.0026 |

## D    COMPARISON TO RANDOM BASELINE

In this section, we show the advantage of using the weakest PCA features in our attack over randomly-picked PCA features. In each experiment, we fix a dataset and a model and run our attack. Figure 7 shows the success rate ($S$) as a function of the number of weak features ($k$). The success rate is the fraction of inputs for which the PCA-guided perturbation (Section 5) resulted in an adversarial example. We compare our approach to a random baseline which generates perturbations using $k$ random PCA features. Results indicate that: (1) our approach succeeds in generating adversarial examples for all models (though smaller models are easier to fool); (2) using weak features has significantly better success rate than randomly selected features: on average 6x higher and up to 32x; (3) in most cases, the higher the $k$, the higher the success rate.

## E    DISTORTION IN THE PCA DOMAIN

In this section, we report the $L_2$ distortion of our attack in the PCA domain. Table 5 summarizes the results for all the benchmarks in Table 2.

