# OpenReview forum: "NETWORK ROBUSTNESS TO PCA PERTURBATIONS"
_ICLR.cc/2021/Conference — Reject_

### Official Review · AnonReviewer3 · 2020-10-26
**This paper proposed a feature perturbation procedure with given feature mappings, which can be used to select robust/weak features and generate adversarial attacks.**

**Rating:** 7
**Confidence:** 4

**Review:**

The overall quality of the paper is good. This paper proposed a feature perturbation procedure, as a comparison to the commonly used perturbation to the original input data. Given access to a feature mapping and a black-box classifier, the proposed procedure is able to select the most robust/weak features. This then can be used for two important tasks: to determine a robust neighborhood for a data point using the robust features and to design adversarial examples using the weak features. For the first task, the feature-based robust neighborhood proposed by this paper is shown by experiments to contain far more points than the traditional input-based neighborhood. For the second task, the feature-based adversarial examples require less query to the black-box classifier and have less distortion from the original data points compared with other competitive methods, and thus are more human-imperceptible. These characteristics make the procedure appealing.
Therefore, the main contribution of the paper (i.e., the perturbation procedure) is important to the ML community and worth further explorations.
The clarity of the paper is good. There is no difficulty in understanding the content and experimental details are provided.

Cons:
1. The overall running time of Alg 1 is a concern.
2. When generating the adversarial examples, a greedy recovery is performed which may be time-consuming when the data dimension is high.
3. The effectiveness of the proposed procedure seems to strongly depend on the feature mappings. The performance under mapping other than PCA is unknown.

Other comments:
1. The experiment showed good results with PCA feature mapping. Are there any other feature mappings that might work well with this proposed procedure (Alg 1)?

---

> ### Author Response · Authors · 2020-11-24
> **Response to the Cons**
>
> Thank you for your kind review and suggestions. We next address your points raised in Cons:
>
>
> (1) While theoretically Alg1 has a high running time, we use it in a practical way by tuning the parameters so that overall, on state-of-the-art models and datasets, it completes within minutes. We note that other attacks also have a one-time preprocessing step (e.g., AutoZoom trains an Autoencoder).
>
>
> (2) Note that our recovery employs an optimization to mitigate the computation time. Further, we show that even with this recovery step, the number of queries we pose to attack ImageNet models is in line with state-of-the-art black-box attacks. Additionally, one may stop the recovery at any point -- we report the L2 distortion of the adversarial examples we generate without the recovery, and it is not too large (Table 2).
>
>
> (3) We are currently exploring other approaches to extract semantic features such as nonlinear PCA and neural network-based approach.

---

### Official Review · AnonReviewer1 · 2020-10-28
**The work proposes a potentially interesting improved black-box attack, but comparison to existing attacks would have to be much more convincing.**

**Rating:** 3
**Confidence:** 4

**Review:**

This work proposes a method to find weak and robust features via sampling, and utilises the method to (a) define feature-guided neighborhoods and (b) to improve score-based black-box adversarial attacks.

I examine the two main contributions (a) and (b) separately. The feature-guided neighourhoods are basically defined as 1D linear subspaces around a given sample x. The subspaces are usually defined along PCA-projections of the data set, although the definitions would also allow non-linear subspaces. The work claims that along some of these directions a classifier D is robust, and that using verification techniques one can show that these robust subspaces can be much larger than simple epsilon-ball neighbourhoods. While this claim might be true, the significance of this insight is unclear to me. Of course, if carefully chosen, one can find large neighbourhoods in which a classifier does not change its decision - but I fail to see what insights can be taken away.

In a second step, this work computes weak features, i.e. directions in the pixel space along which the classifier D is generally susceptible, and uses this to perform a score-based adversarial attack. The work claims higher robustness with fewer iterations than AutoZOOM and GenAttack. This is indeed an interesting direction, but the work misses a lot of relevant prior work on this area that should be discussed and compared against (in particular many other score-based but also decision-based attacks, one would need to take into account the number of queries needed to find the weak PCA directions to compare against other attacks, etc.).

It might be beneficial to remove the discussion of the neighbourhoods and to concentrate on the improvement of black-box attacks (which would have be shown much more convincingly).

---

> ### Author Response · Authors · 2020-11-24
> **Explanations about the significance of our insights and comparison to existing attacks**
>
> Thank you for the review and the suggestions. We next address your questions and concerns.
>
>
> Q: What is the significance of PCA-guided neighborhoods?
>
> A: Our robustness neighborhoods which are defined using *semantic, human-understandable features* make the network robustness more transparent to the user (for example, Figure 1 shows the network is robust to sky color perturbations). To emphasize this, we added an illustration of the meaning of the features mentioned in Figure 1 and 2 (Figure 1(h) and Figure 2(f)) and a short description of the features in Table 1 (Figure 4).
>
> By showing that we obtain larger neighborhoods than distance-based neighorboods, we show that our definition is better suited to interpret the network robustness to perturbations. By the way, we are not the first to identify the need in studying non-distance based neighborhoods, see for example, Fundamental Tradeoffs between Invariance and Sensitivity to Adversarial Perturbations, Tramèr et al., ICML 2020.
>
> Q: Related work is missing.
>
>  A: We present a practical, black-box attack and thus compare to state-of-the-art practical black-box attacks - AutoZoom and GenAttack. In Related Work, we discuss popular attacks. If there is another state-of-the-art black-box attack or another attack which the reviewer believes we should include, we would be happy to add and compare to. We note that the first reviewer requested a comparison to C&W and we added it (Table 2).
>
>
> Q: Why is the number of queries required to compute the weak PCA features counted separately?
>
>  A: Computing the most robust and weakest features is a one-time computation per model, after which we can compute attacks for as many inputs as we want. Thus, we report the amortized number of queries. It is a common practice to separate between preprocessing steps and the attack itself (e.g., AutoZoom trains an autoencoder, and does not consider it when reporting the attack results).
>
> Q:  Why not focus only on the attack?
>
> A: We believe part of the strength of our approach is to show that the same kind of features may be used to prove robustness (if the network is robust to them) or fool the network (if the network is weak with respect to them).
>
> Modifications to the paper relevant to this review: Figure 1(h), Figure 2(f), Figure 4, Table 2 (C&W), and Related Work.

---

### Official Review · AnonReviewer2 · 2020-10-28
**An interesting direction, requires further investigation**

**Rating:** 3
**Confidence:** 5

**Review:**

The authors study the problem of adversarial robustness, aiming to find regions of the input space for which a classifier is robust. Instead of the standard approach of defining a neighborhood around each data point based on some $\ell_p$-norm, they use PCA to identify directions along which the model is robust or brittle. They then use these methods to identify large regions of input space for which models are robust and, in a complementary direction, to craft imperceptible adversarial examples with few model queries.

In general, understanding the set of perturbations that our models are robust to is an important research question. Unfortunately, the current paper does not go into significant depth.

**Robustness neighborhoods.** Τhe input robustness regions computed using this approach are rather unintuitive. What insight is gained by reading Table 1? Do these robustness regions correspond to something concrete and meaningful (e.g., diagonal stripes, brightness) that we can convey to human users of the model? What fundamentally new understanding do we obtain via this analysis?

**New adversarial attacks.**  The space of existing adversarial attacks is huge. By now, there exist so many different approaches for computing examples that fool models. From that perspective, it is not clear what the new attack proposed offers. Similarly to the point above, since these directions do not necessarily capture something human-understandable it is unclear how they reveal a fundamentally new model vulnerability.

Overall, while the research direction is interesting, further exploration is needed to reach novel insights about these models.

Other comments (not affecting score):
- The large numbers representing "number of images" are somewhat misleading. How can we quantify whether these images are actually distinct. I do not think that these number convey significant information and I would thus recommend removing them.
- https://arxiv.org/abs/1807.04200 and https://distill.pub/2019/advex-bugs-discussion/response-3/ also study model robustness to PCA-based perturbations and might thus be worth discussing.

====== POST-RESPONSE UPDATE ======

I appreciate the author's response and the additional illustrations provided. At the same time, my concerns remain:
- I still disagree with the claim that PCA directions are "semantic and meaningful." Yes, some of them might correspond to image changes that are intuitive, e.g., Figure 4, but it is still impossible to draw any such conclusions without manually inspecting individual directions. In other words, what do I learn about my model by reading Table 1?
- I understand the process of counting the number of images. However, I still think that it is a fundamentally flawed metric. Based on this definition of "distinct", if I change the value of a single pixel by 1/255 I get a distinct image. This is clearly not an intuitive behavior. As a model designer, what do I understand about my model by look at these astronomical numbers.

Overall, while I still find the broad direction interesting, I believe that the paper has fundamental issues and is hence unsuitable for publication.

---

> ### Author Response · Authors · 2020-11-24
> **Further explanations about the paper’s key insights**
>
> Thank you for the review and the suggestions. We next address your questions and concerns.
>
>
> Q: Do the robustness neighborhoods correspond to concrete and meaningful perturbations?
>
> A: Yes, our robustness neighborhoods consist of inputs differing with respect to *semantic features* and are thus meaningful and human interpretable. Figure 1 and 2 illustrate the kind of features we use to define our neighborhoods (sky color and lighting conditions, respectively).  To emphasize this, we added an illustration of the meaning of the features mentioned in Figure 1 and 2 (Figure 1(h) and Figure 2(f)) and a short description of the features in Table 1 (Figure 4).
>
>
> Q: What is the advantage of your attack? Do the attack directions capture something human-understandable?
>
> A: First, our black-box attack requires fewer queries and distortion compared to state-of-the-art practical black-box attacks. Second, our attack exposes a new vulnerability of models to small perturbations of weak PCA features. Although the PCA features often capture meaningful features, we do not believe this is essential for attacks, as the goal of attacks is to find imperceivable perturbations that fool models and not human-understandable perturbations.
>
>
> Q: How can we quantify the number of distinct images in a neighborhood? Why using this metric?
>
> A: The number of images in neighborhoods is well-defined mathematically and in particular counts *distinct images*. We provide here the definition. Given a neighborhood defined around an input X with N pixels (dimensions), where the pixel i ranges over $[X_{down}(i), X_{up}(i)]$ and each pixel domain is [0,1], the number of distinct images in the neighborhood is:  $\prod_{i=1}^{N}{1+\lfloor{255\ast(X_{up}(i)-X(i))}\rfloor+\lfloor{255\ast(X(i)-X_{down}(i))}\rfloor}$.
>
> This formula reflects the transition from a continuous range [0,1] to the discrete pixel representation 1-255.
>
> For example, assume the following neighborhood around $X: [[X(1), X(1)+1/255] ,[X(2),X(2)+1/255]]$ (i.e., the neighborhood consists of images with two pixels, each pixel can have one of two possible discrete values). The neighborhood has 4 distinct images:   $[X(1),X(2)],   [X(1)+1/255,X(2)]$,    $[X(1),X(2)+1/255]$,     $[X(2)1/255,X(2)+1/255]$.
>
> To compare our neighborhoods with distance-based neighborhoods, we must use a metric which can quantify both kinds. To this end, we count the number of images. This metric is independent of the neighborhood shape, yet is capable of truly reflecting the neighborhood size. We believe that counting the number of images is better suited to measure neighborhood sizes. For example, reporting that we proved a distance-based neighborhood with epsilon=0.005 (as we have in Table 1) might mislead one to think this neighborhood is larger than one with epsilon=0.0000001. In reality, both neighborhoods contain a single image and are thus practically identical.
>
> Q: Two references are missing.
>
> A: Thanks for the references, we added them to Related Work. The first work (https://arxiv.org/abs/1807.04200) studies classification quality versus vulnerability to adversarial attacks through principal directions and curvatures of the decision boundary.
> The second paper (https://distill.pub/2019/advex-bugs-discussion/response-3) is a followup on Ilyas et al. (2019) (discussed in our paper). This work considers correlation between features and dataset labels (that is, the model is unknown). This setting is different from our black-box setting (see https://arxiv.org/pdf/1804.00097.pdf)
>
>
> Modifications to the paper relevant to this review: Figure 1(h), Figure 2(f), Figure 4, and Related Work.

---

### Official Review · AnonReviewer4 · 2020-10-29
**Confusing setup and experiments**

**Rating:** 4
**Confidence:** 4

**Review:**

The present paper proposes to consider features derived from PCA for the purposes of adversarial attack and defense. They argue that, based on these features, they can verify larger neighborhoods and provide stronger attacks. The neighborhoods are based on the ERAN verifier and the attacks are in comparison to a recent attack called AutoZoom. Defense performance is measured in number of images in the neighborhood, and attack performance in terms of L2 distance.

I was, in general, confused by many of the choices in the paper. The paper proposes to use PCA features, but then measure attack in L2 distance; if the argument is that PCA is a better basis, then why use the pixel-space L2 metric? It seems this could only help to the extent that methods such as projected gradient descent (PGD) are failing to find the optimal L2 attack. Measuring defense capability in terms of number of images also seems like an odd choice, as usually we choose some perceptually meaningful norm and measure according to that. I had also not been familiar with the ERAN verifier prior to this paper, despite being an expert on neural network verification, and it isn't obvious to me whether we should consider it to be competitive with other SDP and LP-based verification methods. Moreover, the PCA features are only used to determine the side lengths of a hyperrectangle that is provided to the verifier, so that this is a fairly indirect test of whether PCA features are useful for defense (indeed, I would like to see a more crisp formulation of what "useful for defense" is meant to mean, since they aren't used to change the model itself).

In the experimental comparison of attacks, standard baselines such as PGD and C&W are missing, such that it is difficult to interpret the results. I checked the AutoZoom paper (the main method compared to) and it also does not compare to PGD, so I feel that the present paper does not provide evidence that the method performs better than baselines.

I think the authors could improve the paper by more crisply articulating its goals, and either using more standard experimental setups and metrics or defending its deviation from these.

== Update after author response ==

Thanks for your response. I believe it is widely agreed upon that black box evaluation is not meaningful in security settings, and that we should use white box attacks. Therefore, I don't find the justification for omitting PGD and CW convincing. I also still do not feel that counting images in pixel space is a meaningful metric. Therefore, I have kept my original score.

---

> ### Author Response · Authors · 2020-11-24
> **Explained setup and experiments**
>
> Thank you for the review and the suggestions. We next address your questions and concerns.
>
> Q: Why report the L2 distortion for the pixel-space and not the PCA-space?
>
> A: The reported L2 distortion is required to properly compare our attack with previous attacks (which have not used the PCA domain). Note that the attack is computed in the PCA domain, and then the perturbation is transformed back to the image domain. However, to address your concern, we added the PCA distortion (Appendix E, Table 5).
>
>
> Q: Why measure the number of images in a neighborhood and not a meaningful norm?
>
> A: Our work defines robustness neighborhoods based on *meaningful semantic features*. To emphasize this, we added an illustration of the meaning of the features in Figure 1 and 2 and a short description of the features in Table 1 (Figure 4).
> By the way, we are not the first to identify the need in studying non-distance based neighborhoods, see for example, Fundamental Tradeoffs between Invariance and Sensitivity to Adversarial Perturbations, Tramèr et al., ICML 2020.
> To compare our neighborhoods with distance-based neighborhoods, we must use a metric which can quantify both kinds. To this end, we count the number of images. This metric is independent of the neighborhood shape, yet is capable of truly reflecting the neighborhood size. We believe that counting the number of images is better suited to measure neighborhood sizes. For example, reporting that we proved a distance-based neighborhood with epsilon=0.005 (as we have in Table 1) might mislead one to think this neighborhood is larger than one with epsilon=0.0000001. In reality, both neighborhoods contain a single image and are thus practically identical.
>
>
> Q: Why ERAN?
>
> A: ERAN is a state-of-the-art analyzer that outperforms LP-based and SDP approaches on many benchmarks -- see the competition results https://sites.google.com/view/vnn20/vnncomp. As far as we are aware, SDP approaches do not scale to large networks (see https://files.sri.inf.ethz.ch/website/papers/neurips19_krelu.pdf, end of page 2).
>
>
> Q: Are PCA features useful for defense?
>
> A: Studying how PCA can be useful for defense is out of scope for our work, but is very interesting as future work. We believe one may employ similar ideas as in Adversarial Training and Provable Defenses: Bridging the Gap, Balunovic, ICLR 2020.
>
>
> Q: Comparison to PGD and C&W is missing.
>
> A: PGD and C&W are white-box attacks, i.e., they assume the attacker has *full access* to the model's weights and architecture, an assumption which is not always true in practice. In contrast, our work is a practical black-box attack where the attacker only has access to the model’s input and output layers. To compare apples to apples, we compare to state-of-the-art practical black-box attacks - AutoZoom and GenAttack. It is a common practice to compare black-box attacks only to black-box attacks (see https://arxiv.org/pdf/1804.00097.pdf). Our results show that our attack computes adversarial examples with fewer queries and less distortion than parallel practical black-box approaches. Nevertheless, to address your concern, we added a comparison to C&W (Table 2) and discussed the difference of white-box attacks and black-box attacks (Related Work).
>
> Modifications to the paper relevant to this review: Figure 1(h), Figure 2(f), Figure 4, Table 2 (C&W column), Appendix E, and the adversarial examples paragraph in Related Work.

---

### Decision · Program_Chairs · 2021-01-07
**Final Decision**

**Decision:**

Reject

**Comment:**

I agree with the majority of reviews that this paper is not sufficiently convincing.